# Prevalence of overweight among Dutch primary school children living in JOGG and non-JOGG areas

**Annita Kobes** *, **Tina Kretschmer, Margaretha C. Timmerman**

Faculty of Behavioural and Social Sciences, Department of Pedagogical and Educational Sciences, University of Groningen, Groningen, The Netherlands

* a.kobes@rug.nl

**Data Availability Statement:** The data underlying the results presented in the study are available from the Dutch Center for Youth Health at the expense of a fee, secretariaat@ncj.nl. The authors of this paper confirm that no special privileges

## Abstract

### Background

One of the most influential integrated approaches towards reducing childhood obesity is EPODE, a program that has been translated to over 20 different countries worldwide.

### Aim

The goal of this study was to explore how JOGG–the Dutch EPODE adaptation–might reduce overweight prevalence among children.

### Methods

To compare whether overweight prevalence was different in JOGG areas vs. non-JOGG areas, in long-term JOGG areas vs. short-term JOGG areas, and in low SES JOGG areas vs. middle/high SES JOGG areas, secondary anthropometric and personal data of 209,565 Dutch children were mapped onto publicly available JOGG data.

### Results

Findings showed that overweight prevalence decreased from 25.17% to 16.08% in JOGG-areas, and from 32.31% to 18.43% in long-term JOGG areas. However, when taking into account SES, the decrease in prevalence was mainly visible in low SES long-term JOGG areas.

### Conclusion

JOGG appeared to be successful in targeting areas where overweight was most prevalent. Low SES areas that had implemented JOGG for a longer period of time, i.e., six years, appeared to be successful in decreasing overweight prevalence.

were received in accessing the data from the Dutch Center for Youth Health that other researchers would not have.

**Funding:** The authors received no specific funding for this work.

**Competing interests:** The authors have declared that no competing interests exist.

## Introduction

Among the most promising childhood obesity interventions are interventions that target overweight across multiple settings (e.g., school, home), levels (e.g., individual, neighborhood), and angles (e.g., diet, physical activity) and are called integrated approaches [1]. One of the most influential integrated approaches is the *Ensemble, Prévenons l'Obésité Des Enfants* (EPODE) program, which was piloted in 2004 in two French communities and has since been introduced in over 20 countries (see here). One of these countries is the Netherlands, which introduced "Jongeren Op Gezond Gewicht" (JOGG, Youth at Healthy Weight) in 2010 [2].

The foundation of EPODE, JOGG, and other adaptations is based on findings of the Fleurbaix Laventie Ville Santé (FLVS) intervention study conducted in the 1990s [3]. Reports on FLVS findings focus predominantly on changes in intervention town inhabitants over time [4–10], whereas comparisons with control town inhabitants have been disseminated to the international scientific community in the form of two publications, suggesting that 1) children in intervention towns had better nutritional knowledge and consumed 6.8% fewer calories per day than children in control towns post-intervention [11], and 2) overweight prevalence was lower in intervention compared to control towns, although the decrease was only observed after eight years [12]. Notably, the prevalence of overweight among families with lower SES was significantly higher in control towns than in intervention towns post-intervention.

Insights from the FLVS study resulted in the design of EPODE, an approach that advocates the installment of stakeholders at two levels: the central level and the local level [3]. At the central level, EPODE advises to organize a Central Coordination Team (CCT) responsible for the program's overall management. Industry partners can show commitment to the EPODE program, but are not meant to intervene in the program's content. At the local level, a project manager activates stakeholders such as school boards, dietitians, and early parenthood consultation clinics to implement EPODE components. Examples of such components are installing water taps at schools, or distributing recipes and storybooks to families to promote healthy food choices. Components can be implemented continuously, or for a specific period of time and can be implemented all at once or in a specific order. Local project managers are free to implement any and all interventions they deem suitable for their community. Thus, which components are implemented differs not only by country, but also between municipalities, cities, and neighborhoods.

EPODE demands substantial investments in terms of money, time, and effort from all stakeholders, which makes their acclaim remarkable given the absence of systematic evidence for the approach's effectiveness. Multiple study protocols have been published [13–15], but only four studies describe the adaptations' effectiveness. An evaluation of the Belgian adaptation showed a trend towards a decrease in overweight ($p = .05$), and overweight + obesity ($p = .06$) in pilot towns compared to the general population [16]. Two other studies–evaluations of OPAL in Australia [17] and TCHP in Spain [18]–showed no statistically significant decreases in overweight prevalence and BMI z-score after 2–3 years and 18 months post-intervention, respectively. The fourth study is another evaluation of the Spanish EPODE adaptation and surveyed children living in intervention areas between 2009 and 2019. During this period of time, overweight and obesity prevalence decreased, however, the systematic effectiveness of the program is difficult to conclude as there was no comparison with a control group [19].

JOGG, the Dutch EPODE adaptation, has been implemented since 2010. Like EPODE, JOGG targets a child's entire community and aims to reduce the prevalence of overweight among children [20], however, no evaluations of its effectiveness have been published in the scientific literature yet.

## Current study

Integrated approaches are promising [1, 21–24], nonetheless, it is meaningful to explore how they might reduce obesity. In this study, we attempt such an exploration using data that are structurally collected among Dutch children in the context of school-based health check-ups. A FLVS study on the change in overweight prevalence suggests that intervention effects were only visible after a long period of time [12], therefore, we take into account trends in overweight prevalence depending on the implementation duration of JOGG.

Given that intervention effects of integrated approaches are often small [23, 24], and the evaluations of Belgian, Australian and Spanish EPODE adaptations resulted in decreasing trends at best [16–18], our hypotheses were conservative. In detail, in order for us to conclude that JOGG is successful, (i) JOGG areas should show a less steep increase in the prevalence of overweight among children than non-JOGG areas. The trend in overweight prevalence among children would thus vary as a function as to whether the area had adopted JOGG or not. Moreover, we expected that (ii) JOGG areas that had implemented the program for a longer period of time (i.e., six years) would show a less steep increase in the prevalence of children with overweight than areas that had adopted the program for a shorter period of time (i.e., three years) or had not adopted the program at all. Thus, the trend in the prevalence of overweight among children was dependent on the duration of the program's implementation.

Children growing up in low SES families are more likely to become overweight/obese [25–27]. FLVS results showed that overweight prevalence among low SES groups was significantly lower in intervention towns than control towns post-intervention [12], and the JOGG approach advises municipalities to focus on low SES neighborhoods [2]. We will visualize exploratorily whether low SES JOGG areas show a different development in overweight prevalence than middle/high SES JOGG areas.

This research's statistical analysis plan was pre-registered on Open Science Framework (here). Modifications to this plan were described in amendments (here).

## Methods

### Ethics approval

The research institute at which this research was conducted does not require ethical approval for secondary data analysis.

### The JOGG program

To implement JOGG, a local government signs a three-year contract with the organization that can be extended for additional three-year periods. JOGG can be implemented in entire municipalities, selected villages or cities, or even in specific neighborhoods. Municipal governments pay the national JOGG organization an annual fee and additionally appoint a local "JOGG-manager" for at least 16 hours per week, whose task is to encourage local institutions in implementing JOGG components, such as realizing healthier breakfasts in family homes, or organizing sports competitions to increase physical activity [28]. The central JOGG organization advises on, but does not carry out these components, which are to be organized and financed by the local governments in addition to paying the annual fee [20]. In other words, JOGG, like EPODE, is an organizational structure that is dependent on substantial investments in time, energy and money by stakeholders. Currently, 40% of Dutch municipalities implement JOGG ($n$ = 143), and JOGG reaches over one million children [2]. JOGG does not structurally evaluate its impact on the level of the individual, which means that children are not usually assessed individually by the JOGG organization. However, if municipalities decide

to collect individual data for evaluation purposes, informed consent of parents is needed. For the present analyses, no such data was used.

## Participants

Dutch children are invited by their local public health service center (GGD) to participate in periodical school-based health check-ups, for which parents receive an information letter via (e-)mail. Parents are furthermore asked to fill in a questionnaire concerning their child's health, and are given the option to opt-out of the health check-up. On the day(s) of the health check-up, a school nurse employed by the local public health service center is present at the school and sees all children individually. Participation in the health check-ups is encouraged, and 25% of GGDs had a response rate of >95% in 2009 [29]. One health check-up takes place in year seven of primary education when children are usually 9–11 years old. A school nurse measures–among other health indicators–children's height and weight. Data are stored at GGDs. Since 2013, many GGDs or affiliated institutions ($N$ = 29) shared data with the Dutch Center for Youth Health (NCJ) (Fig 1). For this study, the NCJ shared data on children's height, weight, sex, and age from 2013–2018.

We received data from 209,571 children. Three entries missed information on the child's sex and were excluded from analyses, resulting in a data set consisting of 209,568 children with 103,776 girls (49.5%) and 105,792 boys (50.5%). 28,083 children were 9 years old (13.4%), 95,635 children were aged 10 (45.6%), 80,453 children were aged 11 (38.4%), and 5,397 children were aged 12 (2.6%). Table 1 shows the sample size per year. We estimated how many of the entire population of Dutch children were likely in year seven of primary education based on data of Statistics Netherlands [30], and compared that number to our sample size (Table 1). Our sample likely contained 5.7% of the population in 2013, 10.9% in 2014, 21.7% in 2015, 26.3% in 2016, 26.6% in 2017, and 17.6% in 2018. Table 1 shows how overweight prevalence in our sample compared to overweight prevalence according to data from Statistics Netherlands.

Group sizes for JOGG/non-JOGG comparisons were generally large, the smallest being the JOGG cohort measured in 2013 ($n$ = 437). The smallest group in the non-JOGG/short-term JOGG/long-term JOGG comparisons was the long-term JOGG cohort measured in 2014

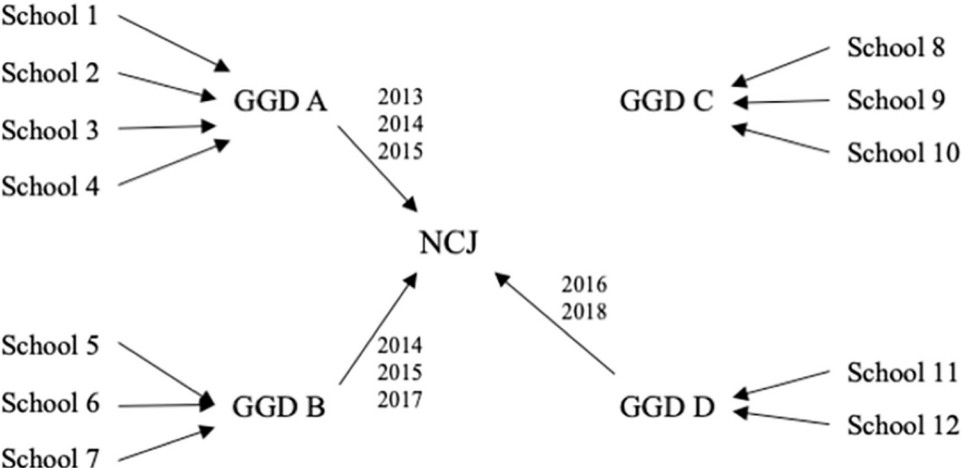

**Fig 1. Graphical display of how data were obtained.** Visual representation of how the anthropometric data was obtained. GGDs send a nurse to schools to measure examine the health of children in year seven of primary education. Some GGDs have shared their data with NCJ (A, B, D) between 2013–2018, but might not have done so for the entire period, e.g., GGD B shared data of 2014, 2015, and 2017 with NCJ.

**Table 1. Representativeness of the sample regarding size and overweight prevalence.**

| | 2013 | 2014 | 2015 | 2016 | 2017 | 2018 |
|---|---|---|---|---|---|---|
| *Number of children in year seven of primary education* | | | | | | |
| Sample N | 11,526 | 21,870 | 42,546 | 50,273 | 50,246 | 33,107 |
| Statistics Netherlands data N* | 202,130 | 200,194 | 196,237 | 191,356 | 188,574 | 187,677 |
| % children in sample | 5.70% | 10.92% | 21.68% | 26.27% | 26.64% | 17.64% |
| *Overweight prevalence among children* | | | | | | |
| Sample prevalence | 14.56% | 13.42% | 15.68% | 16.78% | 15.14% | 14.32% |
| Statistics Netherlands prevalence** | 12.2% | 11.6% | 12.2% | 11.9% | 13.3% | 12.3% |

*This number is an estimation of the total number of Dutch children in year seven of primary education based on the composition of children's age in year seven in our sample. Based on our sample, 13.4% of children were 9 years old, 45.6% were 10 years old, 38.4% were 11 years old, and 2.6% were 12 years old. Accordingly, we calculated the total number of children in year seven in the Netherlands by taking 13.4% of all Dutch 9-year-olds, 45.6% of 10-year-olds, et cetera.

**Statistics Netherlands surveys annually among the Dutch population to provide an overview of the population's health and health behavior. The percentages reported here represent the percentage of Dutch children aged 4–12 with overweight. Statistics Netherlands provides overweight prevalence data accurate to one decimal.

($n$ = 155). The middle/high SES JOGG cohort in 2013 was too small to take into account in statistical analyses ($n$ = 5), as were the middle/high SES long-term JOGG cohorts measured in 2013 ($n$ = 1), and 2014 ($n$ = 2). S1 Table shows all sample sizes per group and year.

## Measures

**BMI.** Height and weight were winsorized at the 97.5th percentile to correct for extreme values [31]. Height, weight and sex were used to compute BMI-for-age and corresponding cut-off scores to indicate weight status; 0 = normal weight, 1 = overweight [32]. Weight status was used instead of continuous BMI, because JOGG aims to reduce the prevalence of children with overweight (i.e., not obesity). Continuous BMI was used as a sensitivity check.

**JOGG implementation status.** The JOGG monitor [2] lists which municipalities implement JOGG, however, some implement JOGG in certain cities within their municipality, or even within specific neighborhoods. All JOGG-managers were contacted to verify the information in the JOGG monitor (response rate = 100%). We registered whether JOGG was implemented before 2013, in 2013, 2014, 2015, 2016, 2017 and 2018. Implementation duration was categorized as follows: non-JOGG areas, short-term JOGG areas (i.e., three years), and long-term JOGG areas (i.e., six years). These 3- and 6-year-periods are contractual periods assigned by the national JOGG organization. We identified JOGG cohorts that continued implementation up to the final measurement in 2018. Three children lived in areas that stopped implementing JOGG after the first cycle and could thus not be organized into a JOGG cohort. These were excluded from further analyses.

**SES.** The Netherlands Institute for Social Research (SCP) ranked each area inhabited by at least one hundred people according to SES between 1998–2017. The SCP's SES-rank is based on average income, percentage of people with low income, percentage of people with low educational attainment, and percentage of unemployed people, and summarized into a factor score [33]. The most recent SES-rank was used to categorize JOGG areas into groups: low SES JOGG areas, consisting of the 25% lowest SES ranks, high SES JOGG areas, consisting of the 25% highest SES ranks, and middle SES JOGG areas, consisting of the remaining 50% SES ranks. Since JOGG–and EPODE–explicitly target low SES groups/areas, it seems sensible to compare low SES group to non-low SES groups, i.e., compare low SES to middle/high SES. Thus, we compared the 25% lowest SES ranks to the 75% middle/highest SES ranks. Table 2 provides an overview of the data frame structure.

**Table 2. Overview of the data frame's structure.**

| ID | Sex | Age | Height | Weight | JOGG <2013 | JOGG 2013 | JOGG 2014 | . . . | JOGG 2018 | SES rank |
|----|-----|-----|--------|--------|-----------|-----------|-----------|-------|-----------|----------|
| 1 | 2 | 11 | 1.34 | 25 | 1 | 1 | 1 | . . . | 1 | 234 |
| 2 | 1 | 10 | 1.19 | 22 | 0 | 0 | 1 | . . . | 1 | 118 |
| n | . . . | . . . | . . . | . . . | . . . | . . . | . . . | . . . | . . . | . . . |

ID, sex, age, height, and weight were provided by the NCJ. JOGG <2013-JOGG 2018 were constructed by AK, and SES rank was extracted from the publicly available information from the SCP. Based on sex, age, height, and weight, we were able to calculate BMI-for-age and corresponding weight status.

## Statistical analysis

To explore whether JOGG areas showed a less steep increase in overweight prevalence among children than non-JOGG areas (hypothesis (i)), we calculated overweight prevalence per cohort. We tested whether there were differences between JOGG and non-JOGG cohorts by means of z-tests, because overweight prevalence per JOGG cohort was expressed in percentages, z-tests were the most appropriate statistical procedure. Overweight prevalence was expressed as a percentage between 0 and 100 and functioned as dependent variable. Whether or not JOGG was implemented functioned as independent variable. Next, to explore whether long-term JOGG areas showed a less steep increase in overweight prevalence than short-term JOGG areas or non-JOGG areas (hypothesis (ii)), we calculated overweight prevalence per cohort. The dependent variable was overweight prevalence among children as a percentage between 0 and 100 and JOGG implementation duration functioned as independent variable.

**Power calculation.** Power was calculated for z-tests comparing JOGG and non-JOGG areas. Effect sizes are expressed in Cohen's $h$, a measure for determining differences between percentages [34]. At $\alpha = .05$ and power of 90%, effect sizes of $h = .16$ (2013), $h = .05$ (2014), $h = .03$ (2015, 2016, 2017), and $h = .04$ (2018) could be detected. For power calculations comparing non-JOGG areas with short-term and long-term JOGG areas, we assumed that all groups were as large as the smallest group. At $\alpha = .05$ and power of 90%, effect sizes of $h = .25$ (2013), $h = .29$ (2014), $h = .06$ (2015), $h = .07$ (2016, 2017, 2018) could be detected.

**Sensitivity analyses.** We conducted sensitivity analyses by combining data of children living in areas that implemented JOGG since 2013 with data of children living in areas that had implemented JOGG *before* 2013, i.e., we extended the long-term JOGG cohort to a *very long-term* JOGG cohort. If long-term JOGG areas truly show a steeper decrease in overweight prevalence, extending the cohort to a very long-term cohort should amplify the results. Furthermore, we conducted sensitivity analyses by altering the dependent variable from a discrete variable, i.e., overweight y/n, to continuous BMI. We conducted t-tests and ANOVAs to test whether BMI significantly differed between cohorts.

**Exploratory analyses.** We re-computed the analyses for hypothesis (i) and hypothesis (ii), while this time distinguishing between low SES areas and middle/high SES areas by means of z-tests.

## Data availaibility

The data of children's height, weight, age, and sex used in this manuscript were provided by the Dutch Center for Youth Health at the expense of a fee. We are contractually obliged to not share these data openly. However, data requests may be sent directly to the NCJ (secretariaat@ncj.nl).

## Results

### Hypothesis 1: JOGG and non-JOGG areas

Overweight prevalence in JOGG areas decreased from 25.17% in 2013 to 16.08% in 2018. Overweight prevalence in non-JOGG areas remained fairly stable with 14.14% in 2013

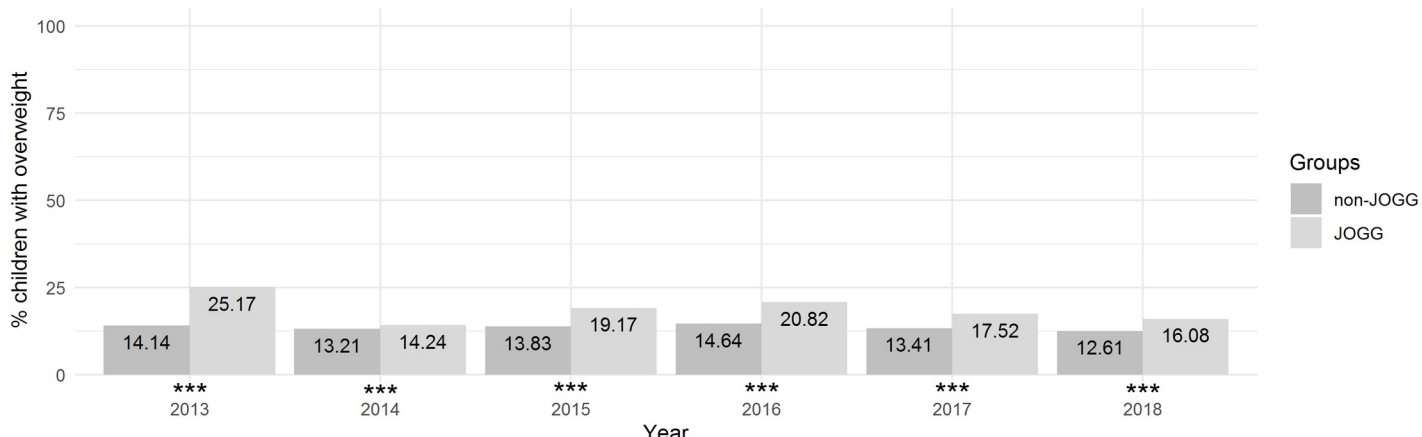

**Fig 2. Prevalence of overweight in JOGG areas and non-JOGG areas.** Visual representation of overweight prevalence among children in non-JOGG areas and JOGG areas between 2013 and 2018. $^{*} p < .05$; $^{**} p < .01$; $^{***} p < .001$.

compared to 12.61% in 2018 (Fig 2). Furthermore, overweight prevalence remained consistently higher in JOGG areas than in non-JOGG areas. The difference in prevalence was largest in 2013 (11.03%) and smallest in 2014 (1.03%). Z-tests indicated significant differences between JOGG and non-JOGG areas in 2013 ($X^2(1) = 41.1, p < .001$), 2015 ($X^2(1) = 208.4, p < .001$), 2016 ($X^2(1) = 311.4, p < .001$), 2017 ($X^2(1) = 160.8, p < .001$), and 2018 ($X^2(1) = 81.4, p < .001$) (Fig 2).

## Hypothesis 2: Non-JOGG, short-term JOGG, and long-term JOGG areas

Overweight prevalence in long-term JOGG areas decreased from 32.31% in 2013 to 18.43% in 2018 (Fig 3). Overweight prevalence among children living in short-term and non-JOGG areas remained fairly stable (Fig 3). Z-tests indicated significant differences between the groups in all years (2013: $X^2(2) = 53.0, p < .001$; 2014: $X^2(2) = 28.7, p < .001$; 2015: $X^2(2) = 12.4, p < .01$; 2016: $X^2(2) = 44.5, p < .001$; 2017: $X^2(2) = 55.5, p < .001$; 2018: $X^2(2) = 74.8, p < .001$). Results of post-hoc analyses are shown in Fig 3 and indicated that, generally, overweight prevalence in non-JOGG areas and long-term areas significantly differed, as did overweight prevalence in short-term and long-term JOGG areas.

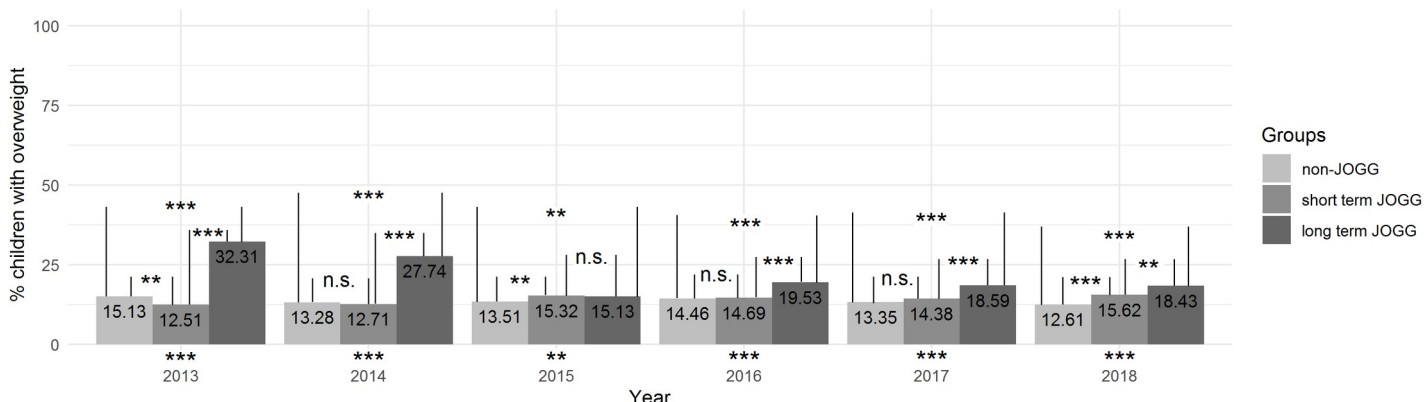

**Fig 3. Prevalence of overweight in long-term JOGG areas, short-term JOGG areas and non-JOGG areas.** Visual representation of overweight prevalence among children in non-JOGG areas, short-term JOGG areas, and long-term JOGG areas between 2013–2018. The statistical significance of the difference between the three groups is expressed below the bars; the results of post-hoc analyses are visualized in the Figure. $^{*} p < .05$; $^{**} p < .01$; $^{***} p < .001$.

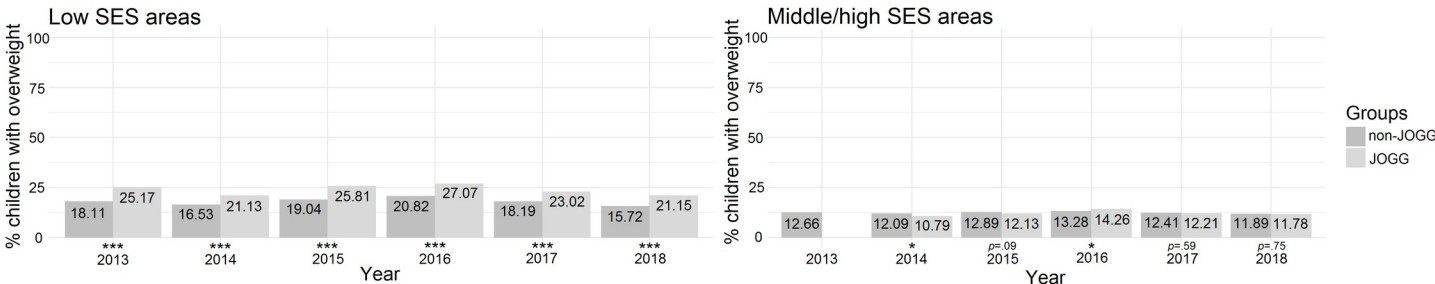

**Fig 4. Prevalence of overweight in JOGG areas and non-JOGG areas, separated by SES.** Visual representation of overweight prevalence in non-JOGG areas and JOGG areas between 2013–2018, separated by SES. On the left, low SES groups are visualized, while middle/high SES groups are visualized on the right. * $p < .05$; ** $p < .01$; *** $p < .001$.

## Sensitivity analyses

We conducted sensitivity analyses to examine whether results would be affected by extending the long-term JOGG cohort, i.e., areas that started implementing JOGG in 2013, with data from areas that implemented JOGG *before 2013*. The decrease in overweight prevalence for long-term JOGG areas visible in Fig 3 disappeared (S1 Fig), implying that extending the 2013 JOGG cohort yielded different results. Furthermore, all analyses were conducted with BMI instead of overweight prevalence as an additional sensitivity check. Patterns of results did not change when altering the dependent variable from overweight prevalence to BMI.

## Exploratory analyses

Exploratory analyses examining the role of SES were conducted. We re-examined hypothesis (i) while distinguishing between overweight prevalence for low SES areas and middle/high SES areas. Fig 4 shows overweight prevalence for low SES JOGG and non-JOGG areas, and middle/high SES JOGG and non-JOGG areas. Overweight prevalence in middle/high SES areas ranged from 10.79–14.26%, while prevalence in low SES areas ranged from 15.72–27.07%. Overweight prevalence in middle/high SES JOGG and non-JOGG areas was fairly equal, while the prevalence in low SES JOGG areas was consistently higher than in low SES non-JOGG areas. What became furthermore apparent, was that the prevalence of 25.17% for JOGG areas in 2013 (Fig 2) consisted entirely of children living in low SES areas (Fig 4).

Next, we re-examined hypothesis (ii) while again distinguishing between low SES and middle/high SES areas. Overweight prevalence in middle/high SES groups appeared to be fairly comparable (Fig 5). However, the overweight prevalence in low SES long-term JOGG areas

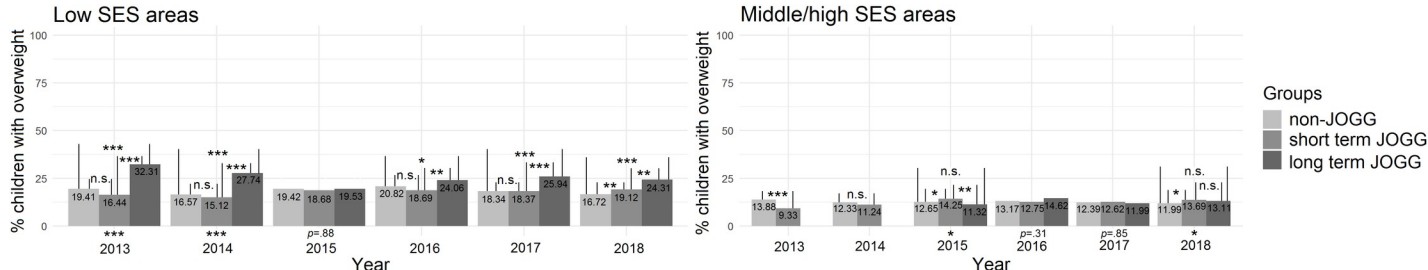

**Fig 5. Prevalence of overweight in short-term JOGG areas, long-term JOGG areas and non-JOGG areas, separated by SES.** Visual representation of overweight prevalence of non-JOGG areas, short-term JOGG areas, and long-term JOGG areas between 2013–2018, separated by SES. On the left, low SES groups are visualized, while middle/high SES groups are visualized on the right. The statistical significance of the difference between the three groups is expressed below the bars; the results of post-hoc analyses are visualized in the Figure. * $p < .05$; ** $p < .01$; *** $p < .001$.

was consistently higher than the prevalence in low-SES non-JOGG and low-SES short-term JOGG areas. The prevalence in long-term JOGG areas in 2013 (= 32.31%) and 2014 (= 27.74%) (Fig 3) consisted entirely of children living in low SES areas (Fig 5).

## Discussion

This study examined variation in overweight prevalence over time in JOGG and non-JOGG areas. We hypothesized that overweight prevalence would overall increase, however, this appeared not to be true. Overweight prevalence in JOGG areas decreased over time compared to non-JOGG areas, and overweight prevalence in long-term JOGG areas decreased over time compared to short-term JOGG, and non-JOGG areas. When the long-term cohort, i.e., six years implementation, was extended to the longer-term cohort, i.e., >6 years implementation, the trend vanished, which might suggest that a longer duration of the program might not necessarily lead to more impact. It might also imply that the duration of implementation in this evaluation has not been long enough to lead to a consistent decrease in overweight prevalence, which would align with results of the longest-term evaluation of EPODE, for which it took eight years to result in a decrease of overweight prevalence [12].

The most remarkable result, perhaps, is that the decreases in overweight prevalence in JOGG areas and long-term JOGG areas are potentially explained by SES, rather than the approach's success: the JOGG cohort in 2013 consisted entirely of children living in low SES areas–where obesity is generally more severe–while the JOGG cohort in 2014 consisted of children living in low SES *and* middle/high SES areas. Similarly, SES could also explain the decrease in overweight prevalence in long-term JOGG areas between 2014 and 2015. When taking SES into account, the low SES long-term JOGG areas form the only cohort in which overweight prevalence decreased.

It is unclear why this decrease is visible in low SES long-term JOGG areas and not in other JOGG areas, and our data contain insufficient information to speculate on the basis of findings. Compared to other EPODE evaluations, our results were surprising. No other EPODE evaluations compared overweight prevalence in low SES and middle/high SES areas, however, a Spanish evaluation showed that overweight/obesity prevalence was higher among public school students, which are generally from lower SES families, than among charter and private schools, which are generally populated by higher SES students [19]. Other EPODE adaptation evaluation studies have controlled for maternal education [18], or a SES measure similar to ours [17]. These studies showed that the proportion of children with a healthy weight did not change over a 2-3-year intervention period in Australia [17], and that the Spanish EPODE adaptation did not significantly affect weight development, obesity incidence, or diet quality and physical activity after 15 months compared to control cities [18]. An explanation for our findings could lie in JOGG's implementation; perhaps municipal governments of low SES JOGG areas apply more funding to combatting childhood obesity, or JOGG managers in these areas install more, or more effective interventions. However, this does not explain why the decrease in overweight prevalence is only visible in low SES *long-term* JOGG areas. It could be that the areas that were first to adopt JOGG had the most severe childhood overweight issues, and were thus most committed to decreasing the issue in their areas. Or, a more wry explanation could be that in areas where the issue is most severe, it is easier to achieve results, just like childhood obesity *treatment* is generally more effective than obesity *prevention* [35].

The results of EPODE evaluations–including the present study–should be interpreted with caution. Previous research has emphasized the difficulty of translating a program-as-intended to the everyday practice [36]. The effectiveness of JOGG could be affected by how components are carried out by stakeholders, which adds a level of subjectivity that could not be

incorporated in this and other evaluations [17, 18]. Furthermore, previous studies argued that evaluations of EPODE adaptations should incorporate longer-term effects [16–18]. While this is the longest-term EPODE evaluation thus far, it still has not exceeded the period of time it took for the FLVS intervention to show an effect, i.e., eight years [12].

The evaluation of integrated approaches such as EPODE and JOGG is still in its infancy, and is complicated by often poor- to moderate-quality monitoring and evaluation [37] due to a lack of motivation, resources, time, and knowledge [38, 39]. Furthermore, target groups are often not thoroughly assessed prior to designing the intervention, which makes it more difficult to reach and engage with the target group and achieve the intended change [37]. Depending on what would be regarded as JOGG's target group, different conclusions would be drawn: JOGG as an approach for all children in all Dutch communities does not appear to have remarkable impact on overweight prevalence, which is in line with previous EPODE adaptation evaluations [16–18]. However, if one would regard JOGG as an approach specifically for communities where the issue is most severe, then this study shows first signs of its success. JOGG appears to be very successful in reaching areas where the issue is most severe, i.e., low SES areas, and appears to be also successful in decreasing the prevalence of children with overweight in these areas.

## Limitations and implications for practice and future research

Data used for these analyses are complex (Fig 1). Many, but not all GGDs shared their data with the NCJ, and which GGDs shared their data differed per year. Therefore, we withheld from applying repeated measures analyses. Furthermore, we cannot draw firm conclusions about the representativeness of the data used for this study. Comparisons of data of Statistics Netherlands and our data gave no reason to believe that our sample is a specific subsample of the population of Dutch children in year seven of primary education (Table 1), however, it might be that some subpopulations were more likely than others to participate in the health check-ups. For example, if children in low SES areas are less likely to participate, and even more so if they have overweight, this might have impacted our results. Table 1 shows a comparison between the overweight prevalence among children in our sample, and the overweight prevalence according to Statistics Netherlands, based on a representative sample of children aged 4–12. The overweight prevalence in our sample is higher, making it less likely that our sample contains an underrepresentation of the number of Dutch children with overweight.

Another limitation of this study is that we were not able to include a variety of factors that might have influenced our results. For example, we could not control for the availability and implementation of other interventions than JOGG, which could mean that families living in non-JOGG areas might have participated in obesity-related interventions other than JOGG. When looking at Fig 5, the decrease of obesity prevalence in long-term low SES JOGG areas is clearly visible, while the pattern of overweight prevalence of long-term low SES non-JOGG areas looks much more stable. Thus, even if a plethora of obesity-related interventions would have been implemented in these areas, one could argue that their impact is not as visible as the impact of JOGG, implying that JOGG might have *additional* value compared to other interventions, which would not seem unreasonable to believe because JOGG, in contrast to many other obesity-related programs, offers an organizational structure instead of a fixed type of intervention.

Future research should further investigate the role of SES. While FLVS findings showed that overweight prevalence among low SES groups was lower in intervention towns than control towns post-intervention, our findings showed contradicting results; overweight prevalence was much higher in low SES JOGG areas than non-JOGG areas. EPODE and its adaptations

intend to focus on low SES groups, and our findings suggest that JOGG might have a different impact on different SES levels. It is not only important to know *how* JOGG affects areas, but also *why* JOGG has a different impact on different areas; are JOGG-managers motivated more strongly in certain JOGG areas, do local governments provide more funding for certain areas than others? Many JOGG municipalities collect BMI data (78%), however, in 2018, evaluations of only 35 JOGG municipalities were known to the JOGG organization, which corresponds to 25% of the total number of JOGG municipalities in 2018 [28]. In the most recent JOGG monitor, JOGG municipalities are advised to evaluate JOGG processes and effects more carefully, because insights in what works why is missing. Future effectiveness research should combine local effectiveness evaluations with local process evaluations to see whether JOGG-as-intended is translated to the everyday practice, and whether or not the effectiveness of JOGG depends on what is implemented and by whom at the local level.

With regards to the evaluation of such complex programs, first steps have been taken to work towards a systematic appraisal of such integrated approaches [37]. Previous studies have shown that decisions pertaining to design and methodology are difficult in evaluating integrated approaches, for example, randomization is often difficult which might introduce the risk of sample bias [40]. Another factor which makes it difficult to conduct a randomized controlled trial, is that the inclusion of large numbers of individuals–which is often the case in the evaluation of integrated approaches–is costly [41]. Alternatives to randomized controlled trials have been suggested, such as pair-matched randomization methods or historical controls, or alternative research designs that may permit more longitudinal analysis, such as extended time series designs [41].

## Conclusion

With respect to this paper's overarching aim–exploring how JOGG might reduce overweight prevalence–we showed that JOGG might work well for a particular subpopulation of children, namely children living in low SES areas where overweight is a severe problem. Taken this and previous research [16–18] together, these studies do not yet provide strong evidence for the effectiveness of EPODE adaptations, assuming their aim is to reach all children in a community. This underscores the need for more research on the effectiveness of EPODE and its adaptations: over 20 countries are investing time, money, and energy into combatting childhood obesity by means of an approach which effectiveness has not yet been scientifically demonstrated enough.

## Supporting information

**S1 Table. Number of children per group and per year.** S1A Table shows the number of children for each group. Overweight prevalence in these groups is visualized in Figs 2 and 3. S1B Table shows the number of children for each group, separated by SES. Overweight prevalence in these groups is visualized in Figs 4 and 5.
(DOCX)

**S1 Fig. Visual representation of overweight prevalence in non-JOGG areas, short-term JOGG areas, and very long-term JOGG areas, i.e., the <2013 and 2013 cohorts combined, between 2013–2018.** The statistical significance of the difference between the three groups is expressed below the bars; the results of post-hoc analyses are visualized in the Figure. $*$ $p<0.05$; $**$ $p<0.01$; $***$ $p<0.001$.
(DOCX)

## Author Contributions

**Conceptualization:** Annita Kobes, Tina Kretschmer, Margaretha C. Timmerman.

**Formal analysis:** Annita Kobes.

**Methodology:** Annita Kobes, Tina Kretschmer.

**Supervision:** Tina Kretschmer, Margaretha C. Timmerman.

**Visualization:** Annita Kobes.

**Writing – original draft:** Annita Kobes.

**Writing – review & editing:** Tina Kretschmer, Margaretha C. Timmerman.

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
