## [Decision Letter · Decision Letter 0]

18 Jun 2021

PONE-D-21-07069

Prevalence of overweight among Dutch primary school children living in JOGG and non-JOGG areas

PLOS ONE

Dear Dr. Kobes,

Thank you for submitting your manuscript to PLOS ONE. After careful consideration, we feel that it has merit but does not fully meet PLOS ONE’s publication criteria as it currently stands. Therefore, we invite you to submit a revised version of the manuscript that addresses the points raised during the review process.

We look forward to receiving your revised manuscript.

Kind regards,

Bidhubhusan Mahapatra, Ph.D.

Academic Editor

PLOS ONE

Journal Requirements:

Additional Editor Comments (if provided):

This is an interesting paper with some useful findings. In addition to the reviewers' comments I have couple of minor suggestions. First, I suggest reducing the length of introduction by moving some of the text as part of study settings under method section. Second, please provide more clarity on the consent process. While the authors were not involved in taking the consent, it is important to document in practice what was followed and which IRB approved the process of data gathering.

Reviewers' comments:

Reviewer's Responses to Questions

**Comments to the Author**

1. Is the manuscript technically sound, and do the data support the conclusions?

Reviewer #1: Partly

Reviewer #2: Yes

2. Has the statistical analysis been performed appropriately and rigorously? 

Reviewer #1: Yes

Reviewer #2: Yes

3. Have the authors made all data underlying the findings in their manuscript fully available?

Reviewer #1: Yes

Reviewer #2: Yes

4. Is the manuscript presented in an intelligible fashion and written in standard English?

Reviewer #1: Yes

Reviewer #2: No

5. Review Comments to the Author

Reviewer #1: I am glad that the article deals with one of the major public health challenges among primary school children in a developed country setting. However, malnutrition among school-going age children does exist in the majority of the countries in the world and growing at a faster pace in some of the countries in Asian and African regions. Therefore, the article and its policy implications place great relevance in reducing overweight among school-going children. However, I would like to seek authors' clarification on my queries and comments are given below;

1. Method section, page 6, line 127-9: Authors mentioned the participation process -"Dutch children are invited by their local public health service center (GGD) to participate in 128 periodical school-based health check-ups. Participation is encouraged, and 25% of GGDs had 129 a response rate of >95% in 2009." How does this recruitment strategy ensure the random participation of children in the study? Again estimating the total number of Dutch children in year seven of primary education based on the composition of children’s age in year seven in our sample would not be reliable unless authors use appropriate sample weights provided in the dataset. If participants' nature is non-random, it would be difficult to obtain unbiased results by using any statistical tool. In such circumstances, statistical techniques bases on matching scores may yield better results.

2. Authors' claim on page 14, line 319-321; the trend of decreasing overweight among JOGG vanished when programme continued for more than six years. However, no concrete explanation was provided for such a finding.

3. How did the study control contamination of information (flow of information from intervention to non-intervention) for such a long period. Because the contamination of such type or from other sources within and across families/communities residing in JOGG and non-JOGG areas and interactions over the period and across the cohorts may potentially influence food habits and lifestyle pattern (exercise and type of games, etc.). How do such possibilities may affect the results in this study?

Therefore, I would recommend authors look into these issues and revise the manuscript to their maximum possible extent.

Reviewer #2: The paper presents evaluation data of a community integrated approach to tackle child obesity. Such data is critically needed to guide future intervention development and direction in the ongoing battle against malnutrition. The paper is very well written, good level of detail and clearly presented.

Minor comments:

1. For figures 2-5, for precision please clarify that the y-axis in underweight only

2. Appreciate the y-axis scale, but suggest that a truncated scale is use to enlarge the histograms to improve readability

3. The discussion is well written. Could the authors add discussion on the evaluation methodology- cluster randomised RCT vs current and newer approaches with respect to how these approaches are appropriate or not for future research.

6. PLOS authors have the option to publish the peer review history of their article (what does this mean?). If published, this will include your full peer review and any attached files.

Reviewer #1: No

Reviewer #2: **Yes: **Shane Norris

---

## [Author Response · Author response to Decision Letter 0]

3 Nov 2021

Responses to editor comments

1. First, I suggest reducing the length of introduction by moving some of the text as part of study settings under method section. 

In response to this comment, we have moved the detailed description of the JOGG program to the Methods section (page 6). 

2. Second, please provide more clarity on the consent process. While the authors were not involved in taking the consent, it is important to document in practice what was followed and which IRB approved the process of data gathering.

In response to this comment, we have added a sentence on the consent process that pertains to the health check-ups (p. 6): “Parents are furthermore asked to fill in a questionnaire concerning their child’s health, and are given the option to opt-out of the health check-up.” We also note for the JOGG program that consent is not usually required for participation (p. 6 “JOGG does not structurally evaluate its impact on the level of the individual, which means that children are not usually assessed individually by the JOGG organization. However, if municipalities decide to collect individual data for evaluation purposes, informed consent of parents is needed. For the present analyses, no such data was used.”). Note that we did not use such individual-level data here. 

Responses to comments by reviewer 1

1. Method section, page 6, line 127-9: Authors mentioned the participation process -"Dutch children are invited by their local public health service center (GGD) to participate in 128 periodical school-based health check-ups. Participation is encouraged, and 25% of GGDs had 129 a response rate of >95% in 2009." How does this recruitment strategy ensure the random participation of children in the study? Again estimating the total number of Dutch children in year seven of primary education based on the composition of children’s age in year seven in our sample would not be reliable unless authors use appropriate sample weights provided in the dataset. If participants' nature is non-random, it would be difficult to obtain unbiased results by using any statistical tool. In such circumstances, statistical techniques bases on matching scores may yield better results

Children were not recruited for the purpose of the evaluation presented here but health check-ups are a regular occurrence in Dutch schools where most children participate. With the data at hand, we cannot draw inferences about whether some children are more likely to participate in the school check-ups than others and whether this is related to living in a JOGG neighbourhood. It might be possible that participation in school check-ups is lower in lower SES neighbourhoods, or where JOGG was also implemented more successfully. We discuss this point in the limitations section on page 17 now: “Furthermore, we cannot draw firm conclusions about the representativeness of the data used for this study. Comparisons of data of Statistics Netherlands and our data gave no reason to believe that our sample is a specific subsample of the population of Dutch children in year seven of primary education (Table 1), however, it might be that some subpopulations were more likely than others to participate in the health check-ups. For example, if children in low SES areas are less likely to participate, and even more so if they have overweight, this might have impacted our results. Table 1 shows a comparison between the overweight prevalence among children in our sample, and the overweight prevalence according to Statistics Netherlands, based on a representative sample of children aged 4-12. The overweight prevalence in our sample is higher, making it less likely that our sample contains an underrepresentation of the number of Dutch children with overweight.”

2. Authors' claim on page 14, line 319-321; the trend of decreasing overweight among JOGG vanished when programme continued for more than six years. However, no concrete explanation was provided for such a finding. 

Reviewer #1 is right: we only mention that this “suggests that program duration does not appear to systematically affect program success”. We have elaborated more on this in the discussion section, see page 15: “When the long-term cohort, i.e., six years implementation, was extended to the longer-term cohort, i.e., >6 years implementation, the trend vanished, which might suggest that a longer duration of the program might not necessarily lead to more impact. It might also imply that the duration of implementation in this evaluation has not been long enough to lead to a consistent decrease in overweight prevalence, which would align with results of the longest-term evaluation of EPODE, for which it took eight years to result in a decrease of overweight prevalence” 

3. How did the study control contamination of information (flow of information from intervention to non-intervention) for such a long period. Because the contamination of such type or from other sources within and across families/communities residing in JOGG and non-JOGG areas and interactions over the period and across the cohorts may potentially influence food habits and lifestyle pattern (exercise and type of games, etc.). How do such possibilities may affect the results in this study?

Therefore, I would recommend authors look into these issues and revise the manuscript to their maximum possible extent.

We are grateful for this comment and have added a section discussing possible contamination of information on page 17/18, which reads as follows: “Another limitation of this study is that we were not able to include a variety of factors that might have influenced our results. For example, we could not control for the availability and implementation of other interventions than JOGG, which could mean that families living in non-JOGG areas might have participated in obesity-related interventions other than JOGG. When looking at Figure 5, the decrease of obesity prevalence in long-term low SES JOGG areas is clearly visible, while the pattern of overweight prevalence of long-term low SES non-JOGG areas looks much more stable. Thus, even if a plethora of obesity-related interventions would have been implemented in these areas, one could argue that their impact is not as visible as the impact of JOGG, implying that JOGG might have additional value compared to other interventions, which would not seem unreasonable to believe because JOGG, in contrast to many other obesity-related programs, offers an organizational structure instead of a fixed type of intervention.” 

Responses to comments by reviewer 2

1. For figures 2-5, for precision please clarify that the y-axis in underweight only.

Please note that we did not study underweight. We are not sure what the reviewer is referring to here so have not made a change in response to this comment.

2. Appreciate the y-axis scale, but suggest that a truncated scale is use to enlarge the histograms to improve readability.

We appreciate the reviewer’s suggestion to improve readability feel that adapting the y-axes might give a distorted impression of the prevalence of overweight (bars would become much higher when y-axis is 0-50% instead of 0-100%). To accommodate this comment, we have enlarged the text in Figures 2-5. We hope that the figures can be read and interpreted more easily now. 

3. Could the authors add discussion on the evaluation methodology- cluster randomised RCT vs current and newer approaches with respect to how these approaches are appropriate or not for future research.

In response to this comment, we have added a discussion on how these approaches would compare to your own (“current”) approach in evaluating programs like JOGG. Please find this addition on page 19: “With regards to the evaluation of such complex programs, first steps have been taken toward a systematic appraisal of such integrated approaches [37]. Previous studies have shown that decisions pertaining to design and methodology are difficult in evaluating integrated approaches, for example, randomization is often difficult which might introduce the risk of sample bias [40]. Another factor which makes it difficult to conduct a randomized controlled trial is that the inclusion of large numbers of individuals – which is often the case in the evaluation of integrated approaches – is costly [41]. Alternatives to randomized controlled trials have been suggested, such as pair-matched randomization methods or historical controls, or alternative research designs that may permit more longitudinal analysis, such as extended time series designs [41].”

---

## [Decision Letter · Decision Letter 1]

2 Dec 2021

Prevalence of overweight among Dutch primary school children living in JOGG and non-JOGG areas

PONE-D-21-07069R1

Dear Dr. Kobes,

We’re pleased to inform you that your manuscript has been judged scientifically suitable for publication and will be formally accepted for publication once it meets all outstanding technical requirements.

Kind regards,

Bidhubhusan Mahapatra, Ph.D.

Academic Editor

PLOS ONE

Additional Editor Comments (optional):

Reviewers' comments:

Reviewer's Responses to Questions

**Comments to the Author**

1. If the authors have adequately addressed your comments raised in a previous round of review and you feel that this manuscript is now acceptable for publication, you may indicate that here to bypass the “Comments to the Author” section, enter your conflict of interest statement in the “Confidential to Editor” section, and submit your "Accept" recommendation.

Reviewer #1: All comments have been addressed

2. Is the manuscript technically sound, and do the data support the conclusions?

Reviewer #1: Yes

3. Has the statistical analysis been performed appropriately and rigorously? 

Reviewer #1: Yes

4. Have the authors made all data underlying the findings in their manuscript fully available?

Reviewer #1: Yes

5. Is the manuscript presented in an intelligible fashion and written in standard English?

Reviewer #1: Yes

6. Review Comments to the Author

Reviewer #1: No further comments or suggestions. All the comments are addressed in the revised version of manuscript.

7. PLOS authors have the option to publish the peer review history of their article (what does this mean?). If published, this will include your full peer review and any attached files.

Reviewer #1: **Yes: **Chander Shekhar, PhD, Professor, Department of Fertility & Social Demography,

---

## [Editor Report · Acceptance letter]

9 Dec 2021

PONE-D-21-07069R1 

Prevalence of overweight among Dutch primary school children living in JOGG and non-JOGG areas 

Dear Dr. Kobes:

I'm pleased to inform you that your manuscript has been deemed suitable for publication in PLOS ONE. Congratulations! Your manuscript is now with our production department. 

Kind regards, 

on behalf of

Dr. Bidhubhusan Mahapatra 

Academic Editor

PLOS ONE